# The Distinctive Activation of Toll-Like Receptor 4 in Human Samples with Sepsis

**DOI:** 10.3390/cells11193020

**Published:** 2022-09-27

**Authors:** Patrick Thon, Katharina Rump, Annika Knorr, Birte Dyck, Dominik Ziehe, Matthias Unterberg, Hartmuth Nowak, Lars Bergmann, Alexander Wolf, Maha Bazzi, Jennifer Orlowski, Marcus Peters, Alexander Zarbock, Thorsten Brenner, Michael Adamzik, Tim Rahmel, Björn Koos

**Affiliations:** 1Klinik für Anästhesiologie, Intensivmedizin und Schmerztherapie, Universitätsklinikum Knappschaftskrankenhaus Bochum, 44892 Bochum, Germany; 2Molekulare Immunologie, Ruhr-University Bochum, 44780 Bochum, Germany; 3Klinik für Anästhesiologie, Operative Intensivmedizin und Schmerztherapie, Universitätsklinikum Münster, 48149 Münster, Germany; 4Department of Anesthesiology and Intensive Care Medicine, University Hospital Essen, University Duisburg-Essen, 45147 Essen, Germany

**Keywords:** sepsis, Toll-Like Receptor 4, proximity ligation assay

## Abstract

Clinical success of Toll-Like receptor-4 (TLR-4) antagonists in sepsis therapy has thus far been lacking. As inhibition of a receptor can only be useful if the receptor is active, stratification of patients with active TLR-4 would be desirable. Our aim was to establish an assay to quantify phosphorylated TLR-4 using the proximity ligation assay (PLA). HEK293 TLR4/MD2/CD14 as well as THP-1 cells were stimulated with LPS and the activation of TLR-4 was measured using the PLA. Furthermore, peripheral blood mononuclear cells (PBMCs) from 25 sepsis patients were used to show the feasibility of this assay in clinical material. Activation of TLR-4 in these samples was compared to the PBMCs of 11 healthy individuals. We could show a transient activation of TLR-4 in both cell lines. Five min after the LPS stimulation, the signal increased 6.7-fold in the HEK293 cells and 4.3-fold in the THP-1 cells. The assay also worked well in the PBMCs of septic patients. Phosphorylation of TLR-4 at study inclusion was 2.9 times higher in septic patients compared to healthy volunteers. To conclude, we established a diagnostic assay that is able to quantify the phosphorylation of TLR-4 in cell culture and in clinical samples of sepsis patients. This makes large-scale stratification of sepsis patients for their TLR-4 activation status possible.

## 1. Introduction

Toll-Like receptors (TLRs) are gatekeepers of the immune system. Currently, there are 10 known human variants (TLR1–TLR10) [1]. TLRs are pattern recognition receptors (PRRs) that bind pathogen-associated molecular patterns (PAMPs) and danger-associated molecular patterns (DAMPs) [1]. TLR4 is responsible for recognizing lipopolysaccharide (LPS), a compound of the outer membrane of gram-negative bacteria [2]. On a molecular level, LPS is bound by LPS-binding protein (LBP) and transported to a complex of TLR4 and myeloid differentiation factor 2 (MD-2) with the participation of cluster of differentiation 14 (CD14) [3,4]. This leads to the secretion of proinflammatory cytokines, reactive oxygen species, antimicrobial peptides, chemokines and acute-phase proteins [4,5]. Upon proper binding of LPS to TLR4, the receptor is phosphorylated and further activates a signaling network consisting of myeloid differentiation primary response 88 (MyD88), various interleukin-1 receptor-associated kinases (IRAKs) and subsequently nuclear factor kappa B (NFκB) dimers [3,4]. Therefore, TLR4 is an attractive therapeutic target for immunological disorders such as sepsis. Sepsis is defined as an acute organ dysfunction caused by the dysregulated immune response to a microbial agent [6]. Currently, sepsis is one of the leading causes of death in industrialized nations [7] with a yearly death toll in the millions [8]. As therapeutic options to specifically treat the dysregulated immune response in sepsis have been largely unsuccessful thus far, the TLR4 inhibitors were viewed as a good approach to reduce the mortality in sepsis. Interestingly, this approach worked well in mouse studies [5]. However, no success could be reported when the TLR4 antagonists were administered in human patients [9]. It is noteworthy that the activation state of the TLR4 receptor was not assessed in these clinical trials prior to the recruitment of septic patients, which is understandable as there is currently no way to measure TLR4 activation in clinical material of sepsis patients. The methods that usually work well in cell culture such as immune-precipitation are too material intensive to be feasible. Furthermore, measuring downstream activation such as phosphorylation of ERK or expression of IL-6 is highly unspecific since both molecules are induced by a range of different receptors [10,11,12,13]. However, since the activation state of the specific receptor is an all-important pre-requisite for an antagonist to work, we wanted to establish an assay based on the proximity ligation assay (PLA, [14]) that can quantify the phosphorylation (i.e., activation) of TLR4 in clinical material.

## 2. Materials and Methods

### 2.1. Cell Culture

The THP-1 cells were cultured in a RPMI-1640 medium (Sigma-Aldrich, Taufkirchen, Germany) supplemented with 10% fetal calf serum (FCS) (Thermo Fisher Scientific, Bremerhaven, Germany) and 1% Penicillin and Streptomycin (Thermo Fisher Scientific). The HEK-293 TLR4/MD2/CD14 cells were cultured in DMEM (PAN-Biotech, Aidenbach, Germany) supplemented with 10% FCS and 1% Penicillin, 1% Streptomycin, 100 µg/mL Hygromycin B (Roche Deutschland, Grenzach-Wyhlen, Germany) and 10 µg/mL Blasticidin (InvivoGen Europe, Toulouse, France). All cells were cultured in a humidified atmosphere at 37 °C with 5% CO_2_. For differentiation, the THP-1 cells were seeded in 12-well plates and stimulated with 10 ng/mL phorbol-12-myristat-13-acetat (PMA) (Sigma-Aldrich) for 48 h. Subsequently, the cells were washed and incubated another 48 h in full RPMI medium before being used for the experiment. For LPS stimulation, the cells were seeded on microscope slides and left to adhere for 12–24 h. Then the cells were stimulated with lipopolysaccharides from *E. coli* O55:B5 (Sigma-Aldrich) (final concentration 1 µg/mL) for the indicated times and fixed using a 4% formaldehyde solution (Sigma-Aldrich).

### 2.2. Patient Recruitment

The samples used in this study were part of the SepsisDataNet.NRW cohort [15]. Patients were prospectively recruited at the Universitätsklinikum Knappschaftskrankenhaus Bochum (DRKS00018871; Ethics-Registration No. 18-6606—BR) and the University Hospital Münster (Ethics-Registration No. 2017-513-b-S) according to the Sepsis-3 definition [6]. In the SepsisDataNet.NRW study biomaterials, serum and peripheral blood mononuclear cells (PBMCs) were collected at day one, day four and day eight after study inclusion. In addition, blood was drawn from self-declared healthy subjects and the PBMCs were isolated (Ethics-Registration No. 21-7278, Ethics committee Ruhr University Bochum).

### 2.3. PBMC Isolation

For this study, 20 septic patients with either a hospital-acquired pneumonia or a peritonitis were identified in the SepsisDataNet.NRW cohort. In addition, five septic patients with severe COVID-19 were selected from the cohort. For isolation of the PBMCs, blood samples from sepsis patients and healthy subjects were analyzed using a Ficoll density gradient centrifugation (GE Healthcare Europe, Freiburg, Germany). The phase containing the PBMCs was collected and washed with PBS. After subsequent lysis of residual erythrocytes, the cells of healthy subjects were seeded in a 12-well plate in an appropriate cell number in RPMI medium and stimulated with LPS as described above. The PBMCs of septic and COVID-19 patients as well as stimulated cells from healthy subjects were directly spun to a microscope slide using a cytospin (Cellspin II, Tharmac, Wiesbaden, Germany) according to the manufacturer’s instructions. The cells were then fixed using a 4% formaldehyde solution, dried and frozen at −80 °C until use.

### 2.4. Measurement of Cytokines in Serum

The serum samples were used to quantify the concentration of thirteen cytokines at the time of recruitment. The LegendPlex Human Inflammation Panel 1 (BioLegend, San Diego, CA, USA) was used according to manufacturer’s instructions. Briefly, the serum samples were incubated with LegendPlex beads for antigen capture, washed and incubated with detection antibodies. Subsequently, the fluorescence was quantified in a flow cytometer (Canto II, BD Biosciences, San Jose, CA, USA). When the recorded concentration of a cytokine was below the lower limit of detection (LOD), the value was set to 0 pg/mL; additionally, if a value was recorded as higher than the upper LOD, it was set to the upper LOD.

### 2.5. Proximity Ligation Assay

The PLA was performed as described previously [16]. In short, the cells were re-hydrated, permeabilized with 0.1% Triton X (Carl Roth, Karlsruhe, Germany) and treated with 1% SDS (Carl Roth, Karlsruhe, Germany). After blocking of unspecific binding sides with Duolink Block (Sigma-Aldrich), primary antibodies were applied. For the cell culture, primary antibodies were diluted 1:200 TLR4 (sc-293072, Santa Cruz Biotechnology, Dallas, TX, USA) and 1:100 pan phospho-tyrosine antibody (p-Tyr-1000, #8954, Cell Signaling Technology, Denver, MA, USA) in antibody diluent (Sigma-Aldrich) and incubated at 4 °C over night. For the PBMCs, the antibodies were diluted 1:50 TLR4 and 1:50 p-Tyr-1000 in antibody diluent and incubated at 4 °C over night. After washing of the unbound primary antibody, the slides were incubated with proximity probes Duolink Mouse PLUS and Duolink Rabbit Minus (Sigma-Aldrich) according to the manufacturer’s instructions. After ligation and amplification using a compaction oligonucleotide described before [16], the slides were imaged in a widefield microscope.

### 2.6. Image Analysis

Image analysis was performed with FIJI and CellProfiler (Stirling, Swain-Bowden et al. 2021). First, the maximal intensity projections were calculated using FIJI and the images compatible for CellProfiler analysis were created. For PBMCs, the DAPI channel was used to identify and mark intact nuclei. Images were analyzed in CellProfiler and subjected to a pipeline containing the following modules: identify primary objects (nuclei), identify secondary objects (cells), enhance or suppress features, identify primary objects (PLA signals) and relate objects (PLA signals to cells). For each condition, a minimum of 50 cells were analyzed with at least three images per condition.

### 2.7. Quantitative RT-PCR

Expression of cytokine mRNA was measured after the LPS stimulation of the cell lines. RNA was extracted using the RNeasy Kit (Qiagen, Hilden, Germany) according to the manufacturer’s instructions. Subsequently, 1 μg of mRNA was transcribed using the High-Capacity cDNA synthesis kit (Thermo Fisher Scientific, Waltham, MA, USA). We then subjected 2 μL cDNA in the qRT PCR analysis using primers against interleukin 8 (forward: GCTCTGTGTGAAGGTGCAGT, reverse: CTCTGCACCCAGTTTTCCTT), interleukin 6 (forward: CCTTCCAAAGATGGCTGAAA, reverse: CAGGGGTGGTTATTGCATCT), interleukin 10 (forward: GTTTTACCTGGAGGAGGTGATG, reverse: TGCCTTTCTCTTGGAGCTTAT) and TNF-α (forward: ATGAGGTACAGGCCCTCTGAT, reverse: CCAGGCAGTCAGATCATCTTC). Normalization was performed using primers against β-actin (forward: CATGTACGTTGCTATCCAGGC, reverse: CTCCTTAATGTCACGCACGAT) and n-folds were calculated using the ΔΔCt method.

### 2.8. Statistics

Statistical analysis was performed using SPSS (IBM, Armonk, NY, USA, Version 28). The PLA signals per cell and other continuous data were compared using a Kruskal–Wallis test for multiple comparisons and a Mann–Whitney U test for pair wise comparisons. Categorical parameters were analyzed using the Fisher Exact test. Correlations were calculated with the Pearson correlation analysis.

## 3. Results

### 3.1. Establishment of the PLA Assay in HEK-293-TLR4/MD2/CD14 Cells

HEK-293 cells overexpressing the TLR4 receptor together with MD2 and CD14 were used to establish the PLA assay. After optimization of the antibody concentration, the antibody pair used and the LPS stimulation, we could show a transient activation of the TLR4 in the HEK-293-TLR4/MD2/CD14 cells upon stimulation with LPS (Figure 1a). At the ground state, the HEK cells showed a mean PLA signal of 1.7 (+/−1.9) signals per cell, which increased after 5 min of 1 μg/mL LPS to 11.4 (+/−6.9) signals per cell (*n* = 3, each *p* < 0.001, Figure 1b). Activation of TLR4 could further be confirmed by an increase in the expression of interleukin 8 (IL-8) after the LPS stimulation to 34-fold (+/−4; *p* < 0.001, Figure 1c).

### 3.2. Validation of the Phospho-TLR4 PLA Assay on Endogenous Protein Expression

As the HEK cells ectopically express TLR4 and its relevant components, we tested the PLA assay on the THP-1 cells, which endogenously express TLR4. Upon stimulation with LPS, we could again show an increase in the phosphorylation level of TLR4 from 0.4 (+/− 0.3) signals per cell (unstimulated) to 1.7 (+/− 0.16) signals per cell (5 min after LPS stimulation, *n* = 3, each *p* < 0.001, Figure 2b). In the THP-1 cells, we could also confirm the activation of the TLR4 signaling cascade using qPCR by showing an increase in interleukin 6 (IL-6) by 4148-fold (+/− 2005, *p* < 0.001) after 24 h of incubation with 1 µg/mL LPS (Figure 2c). Undifferentiated THP-1 cells that retain the monocytic state reacted slower with a maximum of TLR4 activation after 15 min of the LPS stimulation (0.8 +/− 0.1 Signals per cell vs. 2.4 +/− 0.3 signals per cell, *n* = 2, data not shown).

### 3.3. Translation of the Assay to Peripheral Blood Mononuclear Cells (PBMCs) of Healthy Individuals

The PBMCs from healthy individuals equally showed an increase in the PLA signals from 0.4 signals per cell (+/− 0.8, unstimulated) to a maximum of 1.6 signals per cell (+/− 0.1, *n* = 3, each *p* < 0.001, Appendix A) after the LPS stimulation. The kinetics of the three individuals differed slightly with a maximum of TLR4 activation between 5 and 15 min after stimulation.

### 3.4. Cohort Description

In order to test the assay in the clinical material, we quantified the TLR4 activation in the PBMCs of 25 critically ill patients. The median age of the cohort was 63 years and 60% were male. The median sequential organ failure assessment (SOFA) score of the cohort at study inclusion was 9. Of the 25 critically ill patients, 20 were diagnosed with a polymicrobial sepsis and five patients suffered from a SARS-CoV-2 induced sepsis. At study inclusion, the median serum concentration was 156 pg/mL (IQR: 531 pg/mL) for IL-6 and 8 pg/mL (IQR: 12 pg/mL) for IL-10. The median C-reactive protein (CRP) was 18 mg/dL and the median procalcitonin (PCT) concentration was 12.2 ng/mL at study inclusion. A more in-depth characterization is shown in Table 1. As a control group, we measured the TLR4 phosphorylation in 11 self-declared healthy individuals. The median age of this cohort was 48 and 36% were male.

### 3.5. TLR4 Activation in Septic Patients

The mean phosphorylation of TLR4 at study inclusion was 2.0 signals per cell (+/− 3.3) in septic patients compared to 0.7 signals per cell (+/− 0.5) in healthy volunteers (*p* = 0.788, Figure 3a,b). Interestingly, with 4.2 signals per cell (+/− 0.9) the COVID-19 patients showed a significantly higher phosphorylation than both the bacterial septic (*p* = 0.017) and the control patients (*p* = 0.002, Figure 3c). In addition, while phosphorylation dropped in the septic patients over the next four days (day four: 1.1 signals per cell; +/− 1.1, Figure 3b) it remained high in the COVID-19 patients (3.9 signals per cell; +/− 1.2). Notably, the variability of the signals per cell was visibly higher in the septic group than in the control group. The coefficient of variation (CV) of the healthy subject group was 79%, which was less than half of the septic group (167%). Interestingly, we found a time-delayed correlation of the TLR4 phosphorylation to IL-6 serum concentrations in the septic patients similar to the cell culture experiments. TLR4 phosphorylation at study inclusion correlated well to IL-6 serum concentration at day four (r = 0.642 *p* = 0.003) and phosphorylation at day four correlated to IL-6 concentration at day eight (r = 0.827 *p* = 0.022). While we saw a small detrimental effect of TLR4 activation (cutoff = 2 signals per cell) on the 30-day survival in the entire cohort (about 10% lower survival in the activated group), it did not rise to the level of significance (*p* = 0.435).

## 4. Discussion

In this work, we could establish an in situ proximity ligation assay able to quantify TLR-4 activation in the cell culture and the clinical material of critically ill patients. TLR4 has long been proposed to be a promising target for immune-modulative sepsis therapy. Evaluation of the activity state of this receptor prior to treatment was thus far not possible. This may be the reason for the failure of clinical trials with the TLR4 antagonists [9]. In our work, we introduced a PLA assay to measure and quantify the activation of TLR4 by examining its phosphorylation in cell culture models and clinical samples of sepsis patients. The combination of a specific receptor antibody with a generic phospho-tyrosine antibody in a PLA assay has been successfully employed before [17]. The transient activation pattern found in all cell lines that we evaluated with a maximum activation between 5 and 15 min is comparable to the results described in the literature [18]. This, as well as the qPCR results showing a stimulation of the immune network subsequent to TLR4 phosphorylation, strengthens our conclusion that the assay accurately captures the activation of TLR4. Interestingly, the LPS stimulation series with the PBMCs from healthy donors showed some inter-individual variability in the time it took to reach maximal activation of the PBMCs. Although we only evaluated three donors, these results already point towards the high heterogeneity we typically see with the immune system. This heterogeneity can also be seen in the cohort of critically ill patients as the CV was notably higher in the sepsis patients than in the control group. Interestingly, we could show that most polymicrobial sepsis patients did not show an elevated activation of TLR4. This helps to explain the failure of the TLR4 antagonists in clinical trials. We could, however, identify a small subgroup of patients with elevated activation of TLR4 for which anti-TLR4 treatment might be warranted. These patients showed no obvious association with the gram status of the underlying infection, as one might have suspected, and can thus far only be identified by measuring the activation of TLR4. However, there are multiple possible confounders. There are often different types of pathogens present in sepsis patients, especially in patients with impaired immunity. In addition, blood cultures do not always identify all bacteria in the sample. Therefore, we can imagine a bias where a gram-negative super- or co-infection was not detected in the clinic. At the same time, a long gram-negative infection might lead to the exhaustion of the signaling potential of the TLR4 and subsequently result in an impaired activation [18,19,20]. We also see no significant association with 30-day survival of the patients. However, as our study was not powered to study survival effects, further work will be needed to evaluate the effect that activation of TLR4 might have in septic patients. Furthermore, a much-needed subgroup analysis of the patients will have to be conducted in larger patient cohorts. Interestingly, a set of patients with severe COVID-19, originally included as a negative control, showed high-activation levels of TLR4 at both time points. It has been proposed recently that SARS-CoV-2 could directly activate TLR4 [21] and this explanation would fit well to our data. However, because critically ill COVID-19 patients often harbor bacterial superinfections [22], we cannot exclude this here. Again, larger studies specifically on COVID-19 are needed to shed more light on this intriguing finding.

## 5. Conclusions

To conclude, in this study we show the feasibility of a PLA assay to study the activation of TLR4 in cell culture and the PBMCs of sepsis patients. We propose that measuring the activation state of this receptor should be a pre-requisite for clinical trials evaluating TLR4 inhibitors. Larger studies are needed in order to determine the extent of elevated TLR4 activation in sepsis patients and the impact of this activation on the outcome.

## Figures and Tables

**Figure 1 cells-11-03020-f001:**
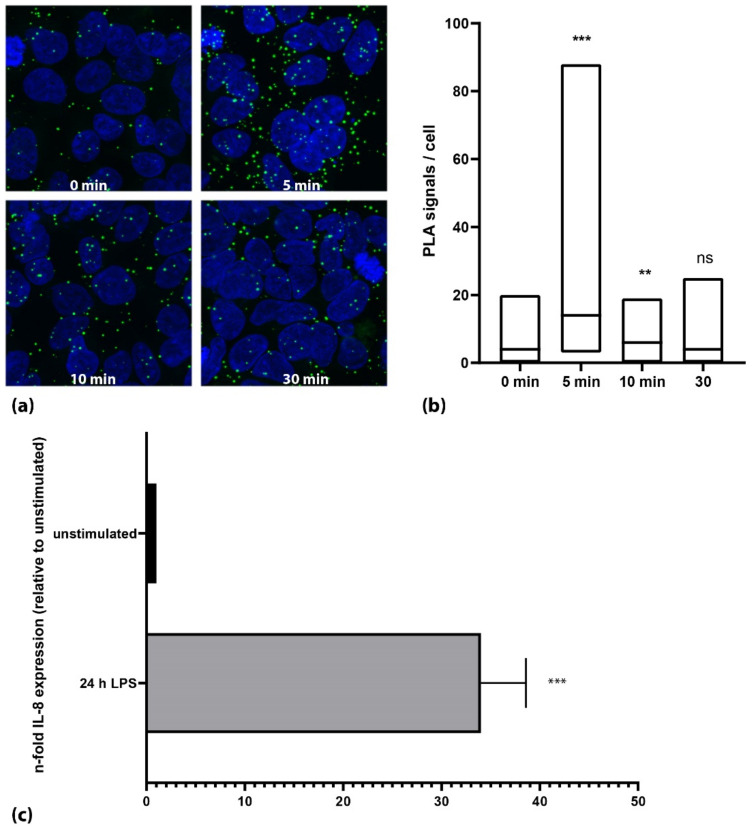
Activation of TLR4 in HEK293-TLR4/MD2/CD14 cells using 1 µg/mL LPS. (**a**) Microscopic images of a representative experiment. Nuclei are shown in blue; PLA signals are shown as green dots. (**b**) Quantification of one representative experiment. The cells showed a marked increase in phosphorylation of TLR4 5 min after stimulation with LPS (*p* < 0.001; Mann–Whitney U test). Activation decreased subsequently to approximately ground state levels. Black lines in the box plots represent the median. (**c**) Stimulation of HEK293-TLR4/MD2/CD14 cells with LPS for 24 h showed a strong increase in IL-8 mRNA expression. (***: *p* < 0.001, **: *p* < 0.01, ns: not significant; *p* > 0.05).

**Figure 2 cells-11-03020-f002:**
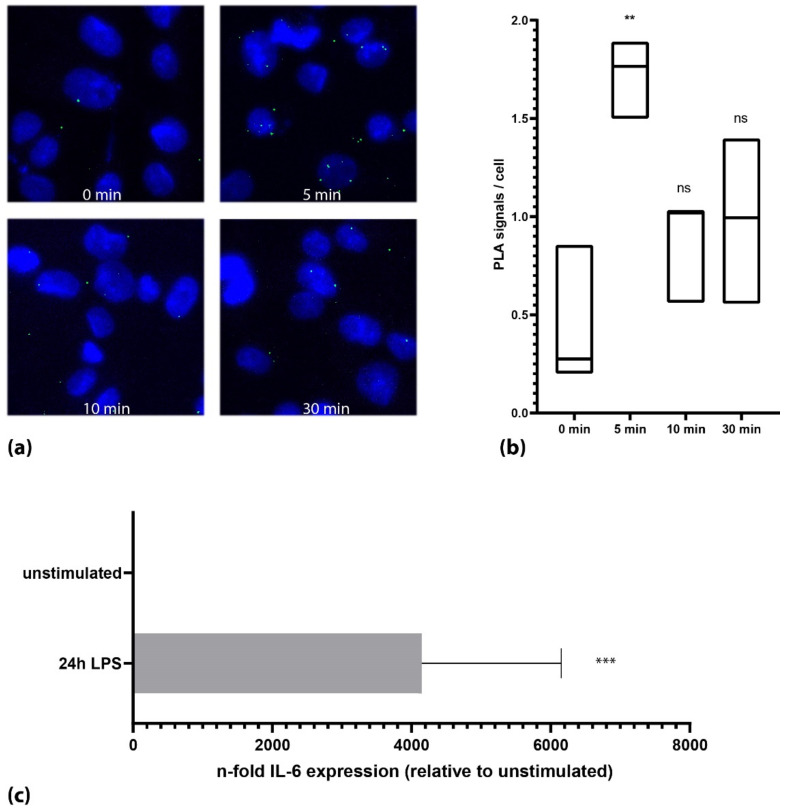
Activation of TLR4 on endogenous expression levels. (**a**) Microscopic images of the THP-1 cells of one representative experiment. Cells showed a transient increase in TLR-4 phosphorylation after 4 min. (**b**) Quantification of all experiments (*n* = 3) together (Mann–Whitney U test). Black lines in the box plots represent the median. (**c**) Stimulation of the THP-1 cells with LPS showed a strong increase in IL-6 after 24 h. (***: *p* < 0.001; **: *p* < 0.01, ns: not significant; *p* > 0.05).

**Figure 3 cells-11-03020-f003:**
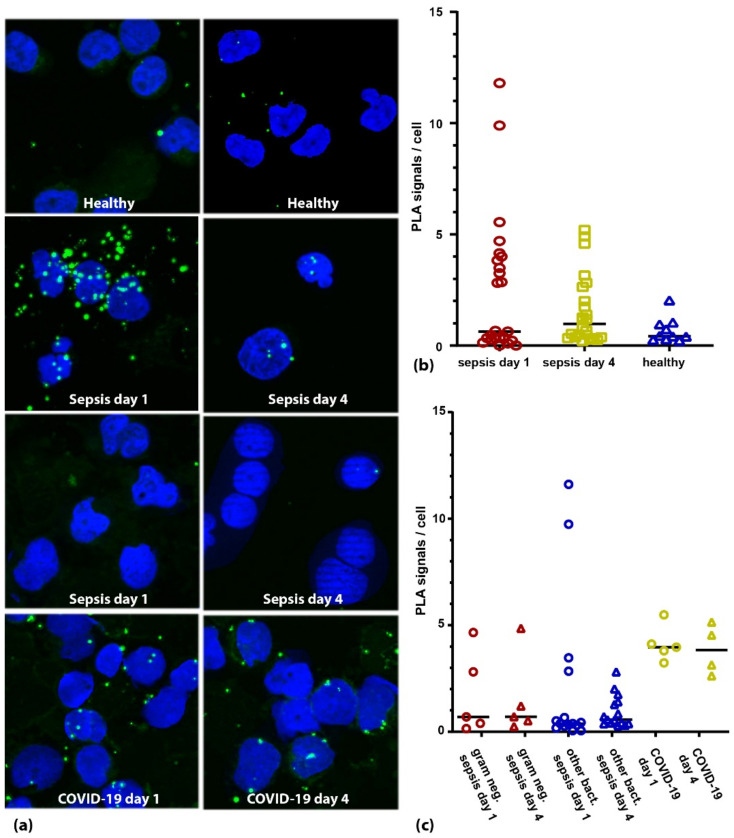
TLR4 activation in PBMCs from sepsis patients. (**a**) Microscopic images of two healthy individuals (**upper panel**), two sepsis patients (panels 2 and 3) and one COVID-19 patient (**lower panel**) at day 1 and day 4 after study inclusion. Nuclei are shown in blue; PLA signals are depicted as green dots. (**b**) Quantification of the TLR4 activation in critically ill patients (bacterial sepsis and COVID-19) at days 1 (red symbols) and 4 (yellow symbols) in comparison to healthy individuals (blue symbols). (**c**) When analyzing the TLR4 activation based on the pathogen of the suspected primary infection, we find no obvious association with gram status (gram negative patients shown in red symbols, other bacterial sepsis patients shown in blue symbols). Strikingly, COVID-19 patients (yellow symbols) showed higher activation than patients with bacterial sepsis both at day 1 and day 4 (*p* = 0.015 and *p* = 0.001 respectively, Mann–Whitney U test).

**Table 1 cells-11-03020-t001:** Base characteristics of 25 critically ill patients included in this study.

n	25
Male gender n (%)	15 (60%)
Age median (IQR)	63 (48–74)
SOFA Score median (IQR)	9 (6–12)
SAPS-2 DRG median (IQR)	37 (24–43)
PCT median (IQR)	12.2 (0.7–22.9)
CRP median (IQR)	18 (11–34)
Lactat median (IQR)	1.3 (0.8–2.1)
pH median (IQR)	7.40 (7.35–7.43)
Kreatinine median (IQR)	2.0 (1.3–3.24)
Focus n (%)	
- Pneumonia	8 (32%)
- Peritonitis	12 (48%)
- COVID-19	5 (20%)
Pathogen	
- Gram-negative bacteria	5 (20%)
- Gram-positive bacteria	2 (8%)
- Mixed	5 (20%)
- Viral	5 (20%)
- unknown	8 (32%)
IL-6 concentration pg/mL median (IQR)	156 (40–571)
IL-10 concentration pg/mL median (IQR)	8 (5–17)
ICU Length of Stay median days (IQR)	8.8 (4.4–22.1)
30-Day Mortality n (%)	8 (32%)

## Data Availability

The data reported herein are available from the corresponding author upon reasonable request.

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
