# Peer review of "The Distinctive Activation of Toll-Like Receptor 4 in Human Samples with Sepsis"

_cells, 2022, doi:10.3390/cells11193020_

Round 1

Reviewer 1 Report

Thon et al developed an assay to quantify TLR-4 activation by the proximity ligation, and demonstrated its feasibility practicability in clinical use in septic patients . The assay is interesting for some clinicians who would like to evaluate TLR4 inhibitors in clinical trials.

The assay developed is novel and original. There wasn't any reliable assay for TLR-4 activation assays can be used for clinical samples.

The manuscript is clearly and concisely presented, easy to understand. Overall, well written.

The conclusion is the assay can be used for clinical samples. From the result from the limited numbers of patient's sample, the assay looks promising for real clinical use. It is addressing the main problem of shortage of such kind of assay for clinical use.

Author Response

R1: Thon et al developed an assay to quantify TLR-4 activation by the proximity ligation, and demonstrated its feasibility practicability in clinical use in septic patients. The assay is interesting for some clinicians who would like to evaluate TLR4 inhibitors in clinical trials.

The assay developed is novel and original. There wasn't any reliable assay for TLR-4 activation assays can be used for clinical samples.

The manuscript is clearly and concisely presented, easy to understand. Overall, well written.

The conclusion is the assay can be used for clinical samples. From the result from the limited numbers of patient's sample, the assay looks promising for real clinical use. It is addressing the main problem of shortage of such kind of assay for clinical use.

A1: We would like to thank the reviewer for reading and evaluating our manuscript. We are pleased that you evaluated it so positively.

Reviewer 2 Report

Comments are attached as pdf document.

Author Response

Reviewer 2:

A brief summary:
Sepsis is systemic damage to organs that is caused by a response to an infection. TLR4 inhibitors are believed to be an important therapeutic target in sepsis’ immune response. While the use of TLR4 antagonist eritoran preserved cardiac function in mice, it was not successful for human patients. The authors hypothesized that the activation state of TLR4 receptor might be different for each patient. Therefore, they aimed to develop an assay to measure the phosphorylated active state of TLR4 in clinical samples from sepsis patients.
They used proximity ligation assay (PLA) on lipopolysaccharide (LPS) stimulated HEK293 TLR4/MD2/CD14 and THP-1 cells to test their hypothesis. Then, they tested their approach in PBMCs from 25 septic patients and 11 healthy controls. After confirming the results reported in this manuscript on large scale, this assay can be used as a diagnostic assay in clinical samples of sepsis patients.

General concept comments
? • The manuscript is well-structured and written with clear and correct English
grammar.

A1: We would like to thank you for investing your time and efforts to improve our manuscript.

? • Twelve out of 18 cited references were not within the last 5 years. Some of them
cannot be replaced by recent citations, however, some can be replaced by most recent publications.

A2: Thank you for pointing this out. We have updated our bibliography. Since the papers we cited before still are very important for this study, we have kept them but supplemented the older ones with new publications re-iterating the cited concepts.

Specific comments
? • Results from HEK293 and THP-1 cell lines are comparable to the results from
primary cells of a small patient cohort. However, to be able to use this assay as a
diagnostic tool, it should be tested in a large cohort.

A3: We agree with the reviewer. This study was to show proof of concept. A larger evaluation of TLR-4 activation in sepsis and COVID-19 patients is planned and will start shortly.

? • Samples are part of the SepsisDataNet.NRW cohort and data will be available upon request.

A4: This is correct.

? • Page 4-Results section
Figure 1b and Figure 2b: Authors reported a mean PLA signal of 1.7 per cell for
unstimulated HEK293 cells and 11.4 after 5 mins of LPS for Figure 1b. Also, PLA signal of 0.4 to 1.7 per cell for Figure 2b. This data is not clear to interpret and understand the results on the bar graph. A Scatter plot like Figure 3b and c will be better.

A5: Thank you for this comment. We have specifically decided to avoid a scatter plot in figure 2b. As PLA signals are either 0 or 1. Therefore in a scatter plot you will have a lot of dots at 0 and many dots at 1 signal per cell and a few at 2. The entire figure looks difficult to interpret. To be consistent we have therefore opted to show a box plot in figure 1 as well (where the higher signal numbers would make a scatter plot more feasible). The differences between the plots and the figures in the text (e.g. 0.4 signals per cell in the text and about 0.3 signals per cell in the plot) stems from the fact that we report means in the text and show medians in the figures. We wanted to give the reader the possibility to see both, to get a better feel for the data distribution. In order to avoid confusion, we have added the fact that the box plot show medians in the figure legends.

? • The authors used n=3 and n=2 (for 15 min LPS stimulation). What are these
numbers? Are they the number of cells that had been analyzed per cell line? If they are, the sample size is not enough to make a strong conclusion. The number of cells that are analyzed should be increased which will result in a smaller SD of the means. Please add which statistical test was used in the figure legend.

A6: We apologize for being unclear. The term n = 3 means that there are 3 independent experiments for this cell line (each with a p < 0.001). The experiment with undifferentiated THP-1 cells was only performed twice (as the expression of TLR4 and hence the activation capability rose significantly after differentiation) and is therefore not included in the manuscript (data not shown). Nonetheless we wanted to share the limited data with the community and therefore included the finding in the text. If the reviewer feels that this is inappropriate we are happy to delete the statement. For each cell culture experimental condition, a minimum of 50 cells were evaluated. In order to avoid confusion, we have clarified this in the methods section of the article. We have also added the statistical test in the figure legends as requested.

? • Authors reported qRT-PCR results in Figures 1c and 2c. However, there is no
information about this experiment in material and methods, such as how many
housekeeping genes have been used for normalization, which primers, etc.

A7: We sincerely apologize for this oversight and thank the reviewer for pointing this out. We have added the section to our methods now.

? • Line 173: Authors reported that they used n=3. Is this number of cells per individual? Have you analyzed cells from all 11 healthy controls?

A8: Thank you for this comment. That section is about LPS stimulation of PBMCs from healthy donors. We have done this for three individuals (hence the n = 3 and the mean of 0.4 signals per cell). After the following section (line 195) we report the mean PLA signal for the 11 healthy donors (0.7 signals per cell).

? • Line 179: Please correct the male gender % from 50% to 60%.
? • Page 6-Line 179, 184, 185: There are inconsistent reported numbers between the
cohort description text and Table 1 such as the median age was reported as 67 in the text and 63 in the table. Similarly, the PCT median and CRP median need to be corrected.

A9: Thank you for your rigorous review. We indeed still had old data in the text (written before the cohort data was complete) and have updated the entire text to match the correct characteristic shown in table 1. We apologize very much for this oversight.

Round 2

Reviewer 2 Report

Thank you for the clarification and changes you made in the manuscript.